# Highly Efficient Antibacterial Polymer Composites Based on Hydrophobic Riboflavin Carbon Polymerized Dots

**DOI:** 10.3390/nano12224070

**Published:** 2022-11-18

**Authors:** Zoran M. Marković, Mária Kováčová, Sanja R. Jeremić, Štefan Nagy, Dušan D. Milivojević, Pavel Kubat, Angela Kleinová, Milica D. Budimir, Marija M. Mojsin, Milena J. Stevanović, Adriana Annušová, Zdeno Špitalský, Biljana M. Todorović Marković

**Affiliations:** 1Vinča Institute of Nuclear Sciences—National Institute of the Republic of Serbia, University of Belgrade, 11000 Belgrade, Serbia; 2Polymer Institute, Slovak Academy of Sciences, Dúbravská Cestá 9, 84541 Bratislava, Slovakia; 3Department of Physical Electronics, Faculty of Science, Masaryk University, Kotlářská 2, 611 37 Brno, Czech Republic; 4Institute of Molecular Genetics and Genetic Engineering, University of Belgrade, Vojvode Stepe 444a, 11042 Belgrade, Serbia; 5Institute of Materials and Machine Mechanics, Slovak Academy of Sciences, Dúbravská Cestá 9/6319, 84513 Bratislava, Slovakia; 6J. Heyrovsky Institute of Physical Chemistry, Academy of Sciences of the Czech Republic, Dolejškova 3, 182 23 Praha, Czech Republic; 7Faculty of Biology, University of Belgrade, Studentski trg 16, 11000 Belgrade, Serbia; 8Serbian Academy of Sciences and Arts, Knez Mihailova 35, 11000 Belgrade, Serbia; 9Department of Multilayers and Nanostructures, Institute of Physics, Slovak Academy of Sciences, Dúbravská Cestá 9, 84541 Bratislava, Slovakia; 10Centre for Advanced Materials Application, Slovak Academy of Sciences, Dúbravská Cesta 9, 84511 Bratislava, Slovakia

**Keywords:** riboflavin, carbon polymerized dots, polymer composites, antibacterial surfaces

## Abstract

Development of new types of antimicrobial coatings is of utmost importance due to increasing problems with pathogen transmission from various infectious surfaces to human beings. In this study, new types of highly potent antimicrobial polyurethane composite films encapsulated by hydrophobic riboflavin-based carbon polymer dots are presented. Detailed structural, optical, antimicrobial, and cytotoxic investigations of these composites were conducted. Low-power blue light triggered the composites to eradicate *Escherichia coli* in 30 min, whereas the same effect toward *Staphylococcus aureus* was reached after 60 min. These composites also show low toxicity against MRC-5 cells. In this way, RF-CPD composites can be used for sterilization of highly touched objects in the healthcare industry.

## 1. Introduction

A major source of pathogen transmission in intensive care units (ICUs) and on hospital premises, including multidrug-resistant isolates, is inanimate surfaces and highly touched equipment (e.g., bedrails, stethoscopes, medical charts, ultrasound machine, fusion pumps, CT scanners, X-ray machines, medical imaging machines, dialysis machines, keyboards, and mobile and stationary telephones). The presence of drug-resistant microbes that can cause nosocomial infection was observed from 40 to 70% of mobile phones of medical professionals working in neonatal ICUs. This finding indicates that there is a serious need for regular phone-cleaning practices to prevent the chances of cross-contamination in hospitals [1]. Any electronic devices used in hospital are recognized as a source of nosocomial infections between hospital personnel and patients [2]. The most common bacteria, isolated from electronic device surfaces, are *Staphylococcus aureus* (*S. aureus*), coagulase-negative *Staphylococcus* (CoNS), *Micrococcus* species, *Pseudomonas* species, and *Escherichia coli* (*E. coli*) [3,4,5].

Traditional hospital sterilization strategies are based on usage of high-level disinfectants: hydrogen peroxide, peracetic acid, and glutaraldehyde; and low-level disinfectants: alcohols, hypochlorites, iodine, and Iodophor [6]. Advanced sterilization technology focuses on chemical-free technology, such as UV rays or gas plasma components. However, there are several disadvantages of both chemical and chemical-free approaches. Firstly, they are toxic to some extent, so medical personnel and patients have to evacuate the premises. Secondly, the sterilization quality is proportional to the human labor invested by the cleaning personnel [7]. An alternative way to mitigate the problem of bacteria colonization is to use photoactive coatings or composites [8,9,10,11,12,13,14]. Highly potent photoactive polymer composites produce reactive oxygen species (ROS) triggered by low-power visible light irradiation from common LED sources. ROS eradicates multidrug-resistant bacteria, quickly disappears, and does not represent a danger to the environment.

Riboflavin (RF), also known as vitamin B2, belongs to the class of water-soluble vitamins. It is not soluble in acetone or chloroform [15]. RF is a very efficient singlet oxygen photosensitizer, with a quantum yield of singlet oxygen generation (Φ_Δ_) = 0.58–0.61 [16,17]. Under UV irradiation, RF produces hydroxyl radicals and superoxide anions [18].

UV or blue light photo-activated water-soluble RF is highly potent antibacterial agent [19]. A study reported extensive eradication of the colony-forming units (CFUs) of *S. aureus*, *P. aeruginosa*, and *S. epidermis* cultured on blood/hematin-agar plates when exposed to UVA/RF. Blue light (λ = 455 nm) triggered photodynamic treatment of *Salmonella*, and its biofilms showed a decrease of 6 log CFU/mL [20]. Highly efficient treatment of bacterial nosocomial infections was demonstrated by Khan et al. [21].

Another study showed that UVA/RF was effective against *S. aureus*, *Pseudomonas aeruginosa* (*P. aeruginosa*), *Staphylococcus epidermidis* (*S. epidermis*), methicillin-resistant *Staphylococcus aureus* (MRSA), multidrug-resistant *P. aeruginosa* (MDRPA), and drug-resistant *Streptococcus pneumonia* (DRSP) by using the Kirby–Bauer method [22]. UV irradiation of RF also suppresses *Candida albicans* fungal biofilms by 24.5% [23]. Blue light photo-illuminated RF inhibits biofilm formation by 34% and 50% of MDR *E. coli*, *MRSA*, and a mix culture, respectively, as compared with the control [24]. *S. aureus* biofilm were eradicated with 92% efficiency using a white light illuminated RF/tertiary amine solution [25]. A polyethylene glycol/RF derivative completely inhibited the growth of *S. aureus*, *P. aeruginosa*, *E. coli*, and *Salmonella* [26].

After photochemical treatment with RB and UV light, *Vesicular Stomatitis Virus* (VSV) and *Herpes Simplex Virus* (HSV) were efficiently eradicated [27]. Two other studies showed that a combination of UV light irradiation and usage of riboflavin eradicated *MERS-Cov* and *SARS-Cov2* viruses [28,29]. 

However, visible light irradiation changes dissolved the RF structure very fast, so it loses capacity to produce singlet oxygen [30]. Anaerobic photodegradation of RF has also been studied in various alcohols and alcohol/water mixtures alone [31]. RF in nanoparticle form is much more resistant to photobleaching and can be used as an anticancer agent [32].

Several studies showed that RF incorporated into chitosan coatings produces OH and H_2_O_2_ radicals, improving their antibacterial properties [33]. The addition of RF led to improved film characteristics, including the thickness, mechanical properties, solubility, and water barrier properties. The CS-RF composite films produced sufficient singlet oxygen under blue LED irradiation for 2 h to inactivate two food-borne pathogens (*Listeria monocytogenes* and *Vibrio parahaemolyticus*) and one spoilage bacteria (*Shewanella baltica*) [34]. Novel composite films of low-density polyethylene (LDPE) containing photosensitizing riboflavin (RF) exhibit antibacterial activity, with reductions of > 99% and 94% in Gram-negative and Gram-positive bacteria, respectively [35]. Poly (vinyl alcohol-co-ethylene) nanofibrous membranes blended with RF showed great photo-induced antibacterial activity against *E. coli* and *L. innocua* in less than 20 min of UVA irradiation [36]. 

In this study, for the first time, highly efficient, pro-oxidant, antibacterial riboflavin-based carbon polymerized dots encapsulated into polyurethane composites are prepared. RF was used as starting precursor to synthesize by bottom-up method carbonized polymer dots (CPDs). CPDs were encapsulated into commercial polyurethane films. CPDs exhibit a unique hybrid polymer/carbon structure, which provides solubility in acetone and chloroform. Namely, carbonized fragments are incorporated into the polymer matrix [37]. In this way, the core of the dots has a high carbonization degree without an obvious crystal structure, whereas the shell exhibits polymer properties [38]. Different reports indicate that CPDs can be divided into two groups, depending on the precursors used: small molecules, such as citric acid, glucose, and amino acids—molecules abundant with –COOH, –OH, or –NH_2_ groups—and polymers [39,40]. The polymer features of these dots are characterized by abundant reactive functional groups, polydispersity, and a highly cross-linked structure [38]. CPDs show high chemical stability, resistance to photo-bleaching and with high potential to use in biomedicine; for example, as an antibacterial agent [41]. 

After encapsulation of CPDs into polyurethane films, the newly developed composites were tested for reactive oxygen species generation, antibacterial activity, and cytotoxicity. These composites showed high potential to generate reactive oxygen species and had strong antibacterial activity, especially toward *E. coli.* On the other hand, the dark cytotoxicity of these polyurethane composites was low.

The main objectives of this study are the following: design of a photodynamic antimicrobial surface based on riboflavin, its characterization, and potential biomedical application.

## 2. Materials and Methods

### 2.1. Materials 

Riboflavin (Carl Roth 97%), ethylenediamine (Carl Roth 99.5%), acetone (Honeywell p.a., Charlotte, NC, USA), fullerene C_60_ (Sigma Aldrich, Rahway, NJ, USA, nicotinic acid (Sigma Aldrich), a nylon membrane filter of 100 nm pore size (Tisch Scientific, Cleves, OH, USA), 2,2,6,6-tetramethylpiperidine (TMP, Sigma Aldrich 99.9%), 5,5-dimethyl-1-pyrroline N-oxide (DMPO, Sigma Aldrich), and Rose Bengal (Carl Roth, Karlsruhe, Germany) were purchased and used as received. Polyurethane films (medical grade) were donated by American Polyfilm Co. (Branford, CT, USA).

### 2.2. Synthesis of RF-CPDs and RF-CPDs Composites

Carbonized polymer dots (RF-CPDs) were prepared by a one-step hydrothermal method. A total of 1000 mg riboflavin and 4 mL ethylenediamine were mixed in 50 mL acetone. At first, the riboflavin was mixed in acetone for 5 min, which resulted in an opaque dispersion. Upon dropwise addition of ethylenediamine, the dispersion got a transparent dark orange color. Then, the mixture was transferred to a Teflon-lined autoclave for heating for 12 h at 180 °C. After the reaction, a dark red product was filtered using a 100 nm nylon membrane filter and centrifuged at 4000 rpm to remove the unreacted riboflavin. The supernatant was collected and used for experimentation. The synthesis route of the RF-CPDs is presented in Figure 1.

The RF-CPD concentration was determined by thermal gravimetry. In total, 10 mL of the RF-CPD colloid was dried, and the mass of deposited film was measured. The concentration was determined as the ratio of the measured mass to volume (10 mL).

The RF-CPD/polyurethane (RF-CPDs/PU) composites were prepared as follows: pieces of polyurethane samples (25 × 25 × 1 mm^3^) were dipped in RF-CPD solution in acetone (50 mL). The concentration of the RF-CPDs was 145 mg/mL. The swelling–shrink–encapsulation method was used to encapsulate the RF-CPDs in PU [10]. The swelling procedure lasted 1 h at room temperature. The RF-CPD composites were dried at 80 °C for 12 h in a vacuum furnace to eliminate acetone from the composites. The same method was used to prepare the C_60_/PU composites as reference samples. The concentration of C_60_ in the toluene solution was 2 mg/mL. The swelling procedure of the polyurethane lasted 12 h at room temperature.

### 2.3. Characterization of the RF-CPDs and RF-CPD Composites

A transmission electron microscope with a probe-corrected FEI/Thermofisher Scientific Titan Themis 300 in scanning mode (STEM) equipped with an energy-dispersive X-ray spectroscopy (EDS) system (Super-X) was used to characterize the surface morphology of the RF-CPDs. The RF-CPDs were dispersed on a TEM support grid. The probe convergence angle was set to 17.5 mrad for imaging applications. Scanning micrographs were acquired simultaneously by 3 detectors: DF2, DF4, and HAADF.

The surface morphology of the RF-CPDs and RF-CPD/PU composites was recorded by AFM (Quesant, New York, NY, USA) operated in tapping mode at room temperature as well. Dispersion of the RF-CPDs was spin coated with mica used as a substrate. Height profiles of more than 200 RF-CPDs and the root-mean-square roughness (RMS) of the neat PU and RF-CPDs/PU composites were determined by Gwyddion 2.53 software [42]. 

### 2.4. Chemical Composition of RF-CPDs

Elemental analysis was performed on a Vario EL III C, H, N, S/O Elemental Analyzer (Elementar GmbH, Langenselbold, Germany) to determine the chemical composition of the RF-CPDs. Samples of the RB-CPDs were dried under reduced pressure and powder samples were used for measurements. The chemical composition of one dot (RF-CPD) was measured by HAADF technique.

XPS analysis of the RF-CPDs nanoparticles was carried out on a SPECS Systems with an XP50M X-ray source for Focus 500 and a PHOIBOS 100 energy analyzer using a monochromatic Al Kα X-ray source (1486.74 eV) at 12.5 kV and 12 mA. Due to the charging effects, the binding-energy axis was referenced to the C1s line at 284.8 eV, assuming that it corresponds to the adventitious carbon. The sample was fixed onto an adhesive copper foil to provide strong mechanical attachment and good electrical contact. The survey XPS spectrum (0–1000 eV BE) was recorded with a constant pass energy of 40 eV, energy step of 0.5 eV, and a dwell time of 0.2 s, while high-resolution XPS spectra of the corresponding lines were taken with a pass energy of 20 eV, energy step of 0.1 eV, and a dwell time of 2 s. The XPS spectra were collected by SpecsLab data analysis software and analyzed using the Casa XPS software package [43]. A standard Shirley background was used for all sample spectra.

The micro attenuated total reflection (ATR) FTIR spectra of the RF-CPD samples deposited on aluminum foil were measured at room temperature in the spectral range from 400 to 4000 cm^−1^ on a Nicolet 8700 spectrometer. The spectral resolution was 4 cm^−1^.

### 2.5. UV-Vis and PL Measurements of RF-CPDs and RF-CPD/PU Composites

UV-Vis spectra of the RF-CPDs and RF-CPD/PU composites were recorded on a Unispec2 LLG spectrophotometer. The measurement range was from 200 to 700 nm. The photoluminescence (PL) spectra of the RF-CPDs, RF-CPD/PU, and RB-CPDs lifetime of all samples were recorded on a Fluorolog spectrofluorometer (Horiba, Kyoto, Japan). UV-Vis and PL measurements were conducted in air at room temperature. The distribution of RF-CPDs in the PU was studied using a confocal Raman microscope (Alpha300 R+, WITec, Ulm, Germany) equipped with a WITec UHTS300 spectrometer (600 lines/mm grating) coupled to an EMCCD camera (DU401A-BR-DD-352, Andor, Abingdon, UK). The sample was excited at a laser wavelength and optical power of 785 nm and 100 µW using a 50× magnification objective (EC Epiplan-Neofluar Dic, NA = 0.8, Zeiss, Oberkochen, Germany) and a 50 mm diameter optical fiber.

### 2.6. Reactive Oxygen Species Generation of RF-CPDs and RF-CPD/PU Composites

#### 2.6.1. Singlet Oxygen Luminescence at 1270 nm Measurements

Time-resolved near-infrared luminescence of O_2_ (^1^Δ_g_) at 1270 nm was observed after excitation by a laser pulse (Nd-YAG laser, wavelength of 355 nm, pulse width ~5 ns, and energy ~1 mJ/pulse) using a homemade detector unit (Ge diode Judson J16–8SP-R05M-HS with amplifier) and averaged 500 times to increase the signal-to-noise ratio. A long-pass filter (λ > 1000 nm) and band-pass interference filters (λ ~ 1270 nm) were placed between the RF-CPD sample and detector. The temporal profiles of the O_2_ (^1^Δ_g_) luminescence recorded after excitation of the absorbance-matched samples were fitted to a single-exponential decay function with the exclusion of the initial part of the plot, which is affected by light scattering and fluorescence. The quantum yields of O_2_ (^1^Δ_g_) formation, Φ_Δ_, were estimated using the comparative method with phenalenone as the standard [44]. 

#### 2.6.2. EPR Measurements RF-CPDs and RF-CPD/PU Composites

The ability of RF-CPDs and RF-CPD/PU to generate singlet oxygen was recorded by electron paramagnetic resonance (EPR) on a Spectrometer MiniScope 300, Magnettech, Berlin, Germany. 2,2,6,6-Tetramethylpiperidine (TMP) was used as a spin trap to determine the singlet oxygen generation. The RF-CPD sample and nicotinic acid, in a concentration of 0.2 mg/mL, in acetone, respectively, were mixed with TEMP (TEMP concentration of 30 mM). The RF-CPD/PU and C_60_/PU samples were dipped in ethanol solution of TEMP (concentration of 30 mM). The measurements were continuously conducted in air 24 h under blue light (BL) irradiation at λ = 470 nm. DMPO (5-dimethyl-1-pyrroline-N-oxide) was used as a spin trap to check the ability of the RF-CPD/PU composites to generate hydroxyl (HO•) and super oxide (O_2_^−•^) radicals. It is known that the DMPO molecules react with hydroxyl and super oxide to form stable spin adducts–radical products with distinct EPR spectra. The RF-CPD/PU composite was dipped in an ethanol solution of DMPO (concentration of 15 mM). The EPR measurements were conducted in air under BL for 24 h.

### 2.7. Photocatalytic Activity of the RF-CPD/PU Composites

For the photocatalytic activity study, the RB-CPD/PU samples were dipped in Rose Bengal (RB, 0.03 mM) water solution and exposed to the blue visible lamp irradiation (470 nm, 3 W). The sample had surface area of 1 × 1 cm^2^ and was dipped in vials with 4 mL of RB solution. To reach absorption and adsorption equilibrium, the solution of RB without the RF-CPD/PU composite was placed in the dark for 1 h before irradiation. The photocatalytic reaction was started when the blue light was turned on. The dye degradation rate was studied by changing the irradiation time (30, 60, 120, and 180 min) and performing the UV-Vis spectroscopy at 549 nm (Unispec2 LLG spectrophotometer) after each measurement. The degradation rate of RB over the RF-CPD/PU composite was determined by the following equation:(1)$$\mathrm{Degradation}\;\mathrm{rate}\;(\%)=\left(C_{0}-\frac{C}{C_{0}}\right)\times 100=\left(A_{0}-\frac{A}{A_{0}}\right)\times 100$$
where *C*_0_ and *C* represent the initial and variable concentrations, whereas *A*_0_ and *A* correspond to initial and variable absorbance, respectively. Photocatalytic experiments were repeated three times.

### 2.8. Contact Angle Measurement of the RF-CPD/PU Composites

The static drop contact angle measurements of the RF-CPD/PU composites were conducted by the Surface Energy Evaluation System (SEE System; Advex Instruments, Brno, Czech Republic) and the software from this system was used for further analysis. The contact angle was measured using 2 μL of deionized water. The accuracy is ±2°, and all measurements were performed in triplicates in the ambient atmosphere and at room temperature.

### 2.9. Antibacterial Testing of the RF-CPD/PU Composites

Antibacterial activity of the RF-CPD/PU composites was tested according to International Standard ISO 22196 (Plastics—Measurement of antibacterial activity on plastic surfaces) using *Escherichia coli* ATCC 1175 and *Staphylococcus aureus* ATCC 25923 [45]. All tests were performed in triplicate, including adequate controls. Samples were prepared as flat squares, 25 × 25 mm^2^, washed with ethanol and UV sterilized for 30 min. Untreated specimens were used for control of bacterial viability, and others materials specimens were incubated under blue light (power 3 W) for 30, 60, and 120 min. Specimens were inoculated with a 0.2 mL bacterial suspension of 5 × 10^5^ cell/mL. Test inoculums were covered with films (20 × 20 mm^2^), and Petri dishes with test specimens were incubated for 24 h at 36 °C. Bacteria were recovered with 10 mL of SCDLP, and different dilutions were plated on NB agar and incubated for 40 to 48 h, after which the bacterial colonies were counted. 

### 2.10. Cytotoxicity Assay of RF-CPD/PU Composites

Cytotoxicity of the RF-CPD/PU composites was investigated by measuring the viability of MRC-5 cells after 24 h of treatment with the tested samples. RF-CPD/PU composites were incubated for 24 h in cell growth medium with 10% fetal bovine serum (0.1 g/mL of medium) at 37 °C with 5% CO_2_. After incubation, the RF-CPD/PU composites were discarded and the incubation medium was diluted with fresh medium to final concentrations of 1%, 10%, 10%, 25%, 50%, and 75%. An undiluted incubation medium was referred to as a concentration of 100%. MRC-5 cells were seeded into a 96-well plate at a concentration of 5 × 10^3^ cells per well and incubated overnight to allow the attachment of the cells to the bottom of the well. The next day, the cells were treated with RF-CPD/PU composites incubation medium. After 24 h of treatment, the cell viability was measured using an MTT assay. Absorbance was measured at 540 nm on a Tecan Infinite 200 PRO microplate reader (Tecan Group, Männedorf, Switzerland). Cell viability was presented as a percentage of the control, which was set to 100%. The control for the RF-CPD-treated cells was untreated MRC-5 cells treated with the vehicle control.

## 3. Results

### 3.1. Solubility of RF-CPDs

The concentration of RF-CPDs was 145 mg/mL, as determined by thermal gravimetry. Such a large concentration indicates that the colloid is stabilized by stearic forces [46]. Upon evaporation of acetone, the RF-CPDs can be fully dispersed in chloroform or toluene.

### 3.2. Surface Morphology of the RF-CPDs and RF-CPD/PU Composites

TEM and AFM were used to visualize the shape and height of the RF-CPDs as well as for determination of their average diameters. Figure 2a presents a large-scale TEM micrograph of the RF-CPD nanoparticles deposited on the TEM support grid. The RF-CPDs have an oblate core-shell structure with an average core diameter of 35.2 ± 2.1 nm, which was determined by statistical analysis of more than 20 micrographs (Figure 2b). The HRTEM micrograph and corresponding electron pattern of the RF-CPDs revealed a paracrystalline carbon structure composed of tiny carbon clusters surrounded by polymer frames (Appendix A). Figure 2c presents the particle size distribution of the RF-CPD nanoparticles. The average diameter of these dots is 57.4 ± 3.1 nm. Figure 2d shows the height distribution of the RF-CPD nanoparticles. The average height is 2.6 nm. Tepliakov et al. reported previously that carbon dots can be treated as polymer-like nanoparticles of amorphous carbon with embedded partially sp^2^-hybridized atomic domains [47]. In this structure, the electrons are partially delocalized over the domain area, but the strong coupling of the amorphous host matrix was maintained continuously. 

Top view AFM images of the surface morphology of the neat PU and RF-CPD composites are presented in Appendix A. The RMS of both samples is 2.31 and 3.47 nm, respectively.

### 3.3. Chemical Composition of RF-CPDs

Elemental analysis revealed the following chemical composition of the RF-CPDs: C 71.23%, N 10.16%, and H 9.66%. The oxygen content was 8.95%. Figure 3b presentsthe HAADF chemical analysis of one RF-CPD, indicated by the black arrow in Figure 3a. The technique of high-angle annular dark-field (HAADF) imaging, which is highly sensitive to atomic-number contrast, reveals that one dot consists of C, N, and O elements, confirming the results of the elementary analysis. 

Apart from the elemental analysis, to further investigate the chemical composition of the RF-CPDs. XPS measurements were conducted. The content of elements detected in the RF-CPD sample by XPS is the following: C = 83.10%, N = 9.62%, and O = 7.27%. All XPS spectra are fitted to study the bonds presented in the RF-CPDs (Figure 3c,d,f). The high-resolution C1s XPS spectrum of the RF-CPDs can be deconvoluted in three peaks at ca. 284.42 eV (sp^2^), ca. 285.40 eV (sp^3^), and at ca. 286.81 eV (C=O) (Figure 3c). These results are also presented Appendix A and showed that the content of sp^3^ bonds is higher, 17.7 At%, compared to the sp^2^ bond. Figure 3d shows the O1s XPS spectrum of the RF-CPD sample and can be fitted only to one peak at ca. 531.68 eV (C=O), indicating the existence of C=O bonds [48]. Figure 3e shows the N1s XPS spectrum of the RF-CPDs and can be fitted to 3 peaks at ca. 399.01 (pyridinic), 400.16 (pyrrolic), and 396.72 eV (C=N-C) [49,50]. The N 1s peak at ca. 396.72 eV corresponds to the sp^2^-hybridized aromatic N atoms (C=N-C) in the aromatic rings.

Figure 3f shows the FTIR spectrum of the RF-CPDs. From this spectrum, the following peaks could be detected: a broad strong band at 3306 cm^−1^, which could be assigned to O–H stretching vibrations in the intermolecular bonding, whereas the peaks at 2952 and 2878 cm^−1^ stem from C–H stretching vibrations of the CH_3_ groups. The peak at 2930 cm^−1^ could be assigned to C–H stretching vibrations of the CH_2_ groups. The peak at 1660 cm^−1^ originated from C=O stretching vibrations, whereas the peak at 1450 cm^−1^ could be assigned to C=C vibrations. The peak at 1365 cm^−1^ stem from a N–O symmetric stretch, whereas the peak at 1298 cm^−1^ could be assigned to C–N stretching of the aromatic amine. The peak at 871 cm^−1^ stems from the N–H primary amine [51,52]. The presence of the functional groups mentioned above indicates the proper intrinsic structure of the RF-CPD—the presence of C=C and C=O bonds with large numbers of hydroxyl groups in the FTIR spectrum [53]. 

### 3.4. Optical Properties of the RF-CPDs and RF-CPD/PU Composites

To study the optical properties of the RF-CPDs, UV-Vis and PL measurements of these dots were performed. In an aqueous solution at pH 7, the UV-Vis absorption spectrum of the riboflavin was characterized by the presence of four bands with maxima at 223, 267, 373, and 444 nm [54]. All bands have high molar extinction coefficients (>10^4^ M^−1^ cm^−1^) that are indicative of π–π* electronic transitions.

Only two bands are present in the UV-Vis absorption spectrum of the diluted RF-CPD acetone colloid (c = 0.072 mg/mL) at 211 and 330 nm (Figure 4a). From this figure, we can observe that the acetone colloid of the RF-CPDs has a strong absorption band at 211 nm, which represents the π–π* transition of C=C. Apart from this peak, there is a very strong peak at 330 nm, corresponding to the n–π* transition of C=O [55]. In the UV-Vis absorption spectrum of the RF-CPD/PU composite, two overlapping bands are dominating at 330 and 357 nm, with a wide shoulder at 412 nm (Figure 4b). The peaks at 330 and 357 nm are due to the n–π* transition of C=O. 

There are various proposed mechanisms related to the PL of carbon dots. All of them includes the following: a quantum core effect due to the π-conjugated domains in the carbon dot core, a defective/or surface state PL due to doping by different heteroatoms, hybridization of the carbon honeycomb and surface functional groups, and PL due to fluorophores and crosslink-enhanced PL emission [56]. One of the important PL features of CPDs is additional PL centers related to the crosslink effect; namely, a polymer-based shell surrounding the carbonized core can contain additional PL centers, i.e., amine groups. If these groups are not constrained into CPDs, the PL centers have weak fluorescence due to strong non-radiative vibrational relaxation [56,57]. In contrast, if they are constrained into CPDs, the vibrational relaxation is suppressed, and the PL intensity increases.

In water at pH 7, riboflavin emits yellow green fluorescence at 520 nm [58]. The decrease in solvent polarity determines a hypochromic shift of the emission band and an increase in the fluorescence quantum yield [59,60]. RF-CPDs emit the strongest yellow red fluorescence at 592 nm (excitation 550 nm-Figure 5a). Figure 5b shows the experimental PL spectra and cumulative PL spectra of the RF-CPDs colloids, which were deconvoluted into two Gaussian fitting peaks: P1 (the so-called intrinsic emission) and P2 (the so-called extrinsic emission), centered at different wavelengths.

The RF-CPD colloid has a narrow P1 peak that represents core emission (FWHM = 21 nm) and a broad stronger P2 peak (FWHM = 71 nm) that originates from defects on the core edges (shell emission) [61]. The P1 and P2 peaks shifted from 504 nm to 617 nm and from 558 nm to 645 nm in the range of excitation, respectively. The integrated area below the P2 peak is 3.6 times bigger than the area below the P1 peak in Figure 5c. The red shift of the PL, depending of the excitation wavelength, indicates that RF-CPDs contain additional PL centers through N–H bonds, which are constrained into the shell of the RF-CPDs. This structure suppresses non-radiative vibrational relaxation and thus PL intensity increases.

RF-CPD/PU composites emit green fluorescence at 509 nm (excitation 450 nm (Figure 5c)). After encapsulation of RF-CPDs into polyurethane, there is a blue shift of PL due to crosslinking of RF-CPDs with polyurethane polymer chains. RF-CPDs display tunable photoluminescence and excitation-wavelength-dependent emission.

The mean value of the fluorescence lifetime of the RF-CPDs was 3.79 ns, with an excitation wavelength of 470 nm. This excitation wavelength was selected to be identical to one used for photodynamic treatment of bacteria.

The CRM mapping revealed homogenous PL across the whole composites without any streaks (Figure 5d). Homogenous PL across the whole sample indicates homogenous encapsulation of the RF-CPDs through the whole polyurethane polymer matrix.

### 3.5. Reactive Oxygen Species Production of RF-CPDs and RF-CPD/PU Composites

It is well known that photoactive nanoparticles, including CDs, produce ROS under certain conditions. The presence of defects and unpaired electrons on the basal plane and edges of the CDs can contribute their ROS generation or quenching. Furthermore, the honeycomb structure of the CDs facilitates charge transfer and electron storage. In our previous investigations, we measured the ROS production by carbon quantum dots (CQDs) produced from citric acid as starting precursors and established that these dots had produced only singlet oxygen. They did not generate hydroxyl radicals or even superoxide anions. Thus, we concluded that only energy transfer was responsible for ROS generation [62].

To check the potentials of new developed RF-CPDs nanoparticles and RF-CPD/PU composites to generate singlet oxygen, we performed two types of measurements: a luminescence method at 1270 nm and EPR. Besides singlet oxygen generation tests, we conducted tests for the production of superoxide anions (O_2_^•−^) by using DMPO as a spin trap. Figure 6a shows the singlet oxygen production of RF-CPDs measured by the luminescence method at 1270 nm. In Table 1, values of the measured quantum yield of singlet oxygen of the RF-CPDs are presented. As could be seen from this table, Φ_Δ_ = 0.28 was measured for RF-CPDs in acetone (Figure 6a). Quenching of singlet oxygen by RF-CPDs is not significant. Less than 10% of the singlet oxygen molecules were quenched by the RF-CPDs themselves. RF-CPDs showed very good photo stability. The UV-Vis spectra of the RF-CPDs before and after experiments were identical. The lifetime of the singlet oxygen was 46.3 µs.

EPR measurements were performed to check the ability of the RF-CPDs colloid and RF-CPD/PU composites to produce singlet oxygen. 2,2,6,6-Tetramethylpiperidine (TEMP) was used as a spin trap. All samples were irradiated with a low-power blue lamp at a wavelength of 470 nm overnight. The EPR spectra presented in Figure 6b show a TEMPO signal produced by the singlet oxygen emitted by the RF-CPDs (red curve) and nicotinic acid (blue curve), with Φ_Δ_ = 0.64. The concentration of both colloids was identical—0.2 mg/mL. 

Figure 6c shows the EPR spectra of PU as a reference (black curve), the RF-CPD/PU composites (red curve), and C_60_ composite (Φ_Δ_ = 1) samples (blue curve). As for the PU sample, it was found that there is no singlet oxygen generation. However, the RF-CQD/PU composites (Abs (470 nm) = 0.98) generate singlet oxygen more than the C_60_/PU (Abs(470 nm) = 0.08) composites. This result is a strong evidence of the presence of the RF-CPDs in the polymer matrix. Figure 6d shows the singlet oxygen production of the RF-CPD/PU composites vs. time. As can be seen from this figure, long-term irradiation did not annihilate the potential of the RF-CPD/PU composites to produce singlet oxygen; i.e., these composites were resistive to photo-bleaching.

Figure 6e shows that RF-CPD/PU composites produce superoxide anions upon blue irradiation for 5 h whereas Figure 6e presents the kinetics of superoxide anions production vs. time. As be seen from this figure, there is an increase in superoxide anions production during 5 h. Thus, light-induced formation of ROS originates from the electron–hole pair; i.e., ROS is generated by photo-excited RF-CPDs via both energy-transfer and electron-transfer pathways [64]. 

### 3.6. Photocatalytic Activity of RF-CPD/PU Composites

The production of ROS induced by both the RF-CPDs and RF-CPD/PU composites prompted us to study whether RF-CPD composites had any photocatalytic activity under blue light irradiation. To monitor the photocatalytic activity of the RF-CPD/PU composites, we have measured the decrease in absorbance maxima of the aqueous solution of the RB dye at 549 nm. Appendix A presents the kinetic rate of the RB dye degradation in the presence of the RF-CPD/PU composite. From this figure, it is observed clearly that the photocatalyst (RF-CPDs/PU composite) irradiated by blue light (λ = 470 nm) degrades 76% of the RB dye solution after 180 min. EPR measurements presented in the previous section have shown that the RB-CPD/PU composites produced both singlet oxygen and superoxide anions. Therefore, we conclude that these radicals are responsible for RB degradation. Namely, under BL irradiation, pairs of electron–hole are created and superoxide anions are formed by accepting electrons via adsorbed oxygen [65], whereas singlet oxygen is formed through electron transfer from RF-CPDs (used as photocatalyst) and environmental oxygen [64]. 

### 3.7. Contact Angle of the PU and RF-CPD/PU Composites

To investigate the wettability of the neat PU and RF-CPD/PU composites, we conducted static drop contact angle measurements. Water droplets were deposited on the RF-CPD/PU composites (Appendix A). It was found that the contact angle for the neat PU sample was 96.15° ± 5.4°, whereas the contact angle for the RF-CPDs/PU composites was 100.8° ± 2.3°. The encapsulation of the RF-CPDs nanoparticles into the polymer matrix has contributed to the hydrophobicity increase of the tested composites slightly. This parameter can affect the antibacterial activity; i.e., the interaction of the bacteria strains with the studied surfaces. 

### 3.8. Antibacterial Testing of RF-CPDs/PU Composites

The antibacterial acting of the CDs is a complex process, including many parameters compared to antibiotics action. CDs are photoactive materials that generate ROS under visible light; but, except for ROS production, degeneration of the cell structure and leakage of the cytoplasm as a result of DNA binding, as well as modulation of gene expression are also responsible for bacterial death [66]. Surface charge is one of the important parameters that affects antibacterial activity as well. Bing et al. reported the antibacterial activity of three types of CQDs with different surface charges [67]. The obtained results showed the following: positively charged CQDs electrostatically interacted with negatively charged *E. coli* and disrupted the bacterial membrane; negatively charged CQDs had a weaker bactericidal effect on *E. coli* compared to positively charged CQDs, whereas the uncharged CQDs did not have any antibacterial activity against *E. coli* and *B. suptilis*. Antibacterial potential of the RF-CPD/PU composites was tested on two types of microbes: *S. aureus* and *E. coli*. Results of the antibacterial testing are summarized in Table 2.

The obtained results showed that the RB-CPD composites had higher efficacy towards *E. coli* compared to *S. aureus*. While antibacterial activity on *E. coli* was detected after 30 min of treatment, *S. aureus* was more resistant. Furthermore, significant bacterial death was detected after 90 min of treatment. The neat PU used as control had no antibacterial activity against the tested bacterial strains. Better sensitivity of the RF-CPD/PU composites against *E. coli* indicates that these dots are probably positively charged and have a better electrostatic interaction with negatively charged *E. coli*. The main roles in bacterial death have the lifetime of singlet oxygen (46.3 µs) and production of singlet oxygen and superoxide anions in high quantities during time (Figure 6d,f). Due to the high oxygen permeability rate in polyurethane, ROS are diffusing outside the polymer pores and kill *E. coli* effectively [68]. They enter the bacterial membrane and cause oxidative stress of the tested bacterial strains. The obtained antibacterial results indicate that these photodynamic, transparent, smooth surfaces can be used as self-sterilizing surfaces in facilities such as hospitals or healthcare institutions. Compared to our previous results, the newly developed composites show better efficacy as an antibacterial agent, especially toward *E. coli*. In our previous research, we used as a precursor to synthesize CDs polyoxyethylene−polyoxypropylene−polyoxyethylene Pluronic 68, and obtained hydrophobic carbon quantum dots were that encapsulated into polyurethane films successfully [10]. 

### 3.9. Cytotoxicity Assay

In the previous section, we established that the newly developed RF-CPD composites are very potent photodynamic antibacterial agents. Thus, it is important to conduct a biocompatibility study to check their toxicity in dark conditions because the low cytotoxicity of an antibacterial agent is one of the mandatory requirements for their usage in biomedicine. Cytotoxic effects of the RF-CPD/PU composites were tested against MRC-5 cells (human lung fibroblasts) using an MTT assay. Lung fibroblasts are very important for maintaining the integrity of the alveolar structure by proliferating and repairing injured areas [69]. MRC5 cells have a normal karyotype and are commonly used for genetic, cytotoxicity, viral infection, and other fibroblast-based assays [70]. Figure 7 presents the cell viability of individual samples in various extract concentrations. The results are presented as a percentage of the control (untreated cells), which was arbitrarily set to 100%. As shown in this Figure 7, control (C-neat PU) and RF-CPD/PU composites did not show any cytotoxic effect, regardless of the extract concentration. Namely, the MRC-5 cell viability was above 70% (the cell viability range was within 76–97% for all concentrations of all the tested samples). 

## 4. Conclusions

In this study, the newly developed RF-CPD/PU composites showed strong antibacterial activity against two types of bacteria strains (*E. coli* and *S. aureus*) after 30 and 60 min, respectively. These composites did not show any cytotoxicity against MRC-5 cells. Green route was used to synthesize these carbonized polymer dots. As a precursor, riboflavin was used. Vitamin B2 (riboflavin) is well known as a strong antibacterial agent against various bacteria and the RF-CPDs showed many features of the precursor used. Namely, the RF-CPDs themselves and RF-CPD/PU composites produce ROS at a high level. The lifetime of the produced singlet oxygen is long, whereas the kinetics of the singlet oxygen and superoxide anions is very good, too. In this way, the newly developed RF-CPD/PU composites can be used as antimicrobial surfaces in healthcare facilities with large number of highly touched objects.

## Figures and Tables

**Figure 1 nanomaterials-12-04070-f001:**
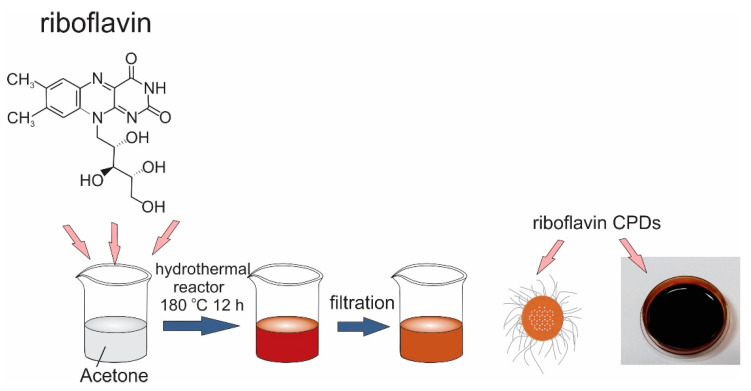
Synthesis route of the RF-CPDs.

**Figure 2 nanomaterials-12-04070-f002:**
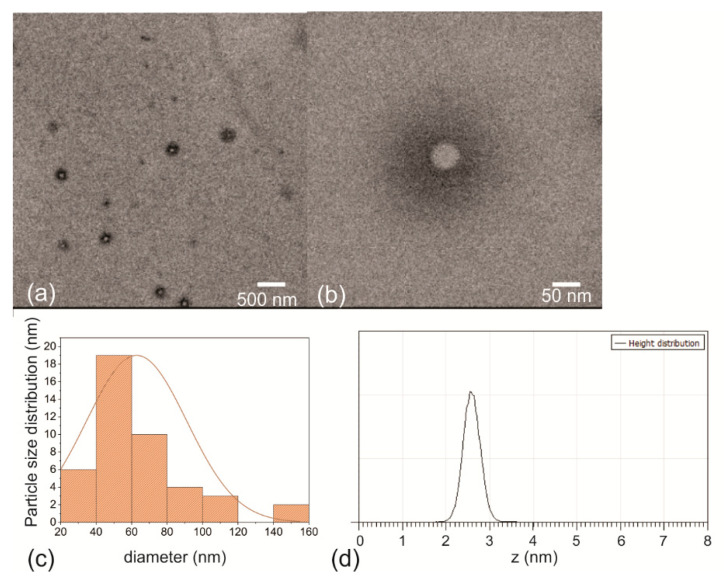
(**a**) Large-scale TEM micrograph of the RF-CPDs (scale 500 nm). (**b**) Small-scale TEM micrograph of a single RF-CPD (scale 50 nm). (**c**) Particle size distribution of the RF-CPD nanoparticles. (**d**) Height profile of the RF-CPD nanoparticles.

**Figure 3 nanomaterials-12-04070-f003:**
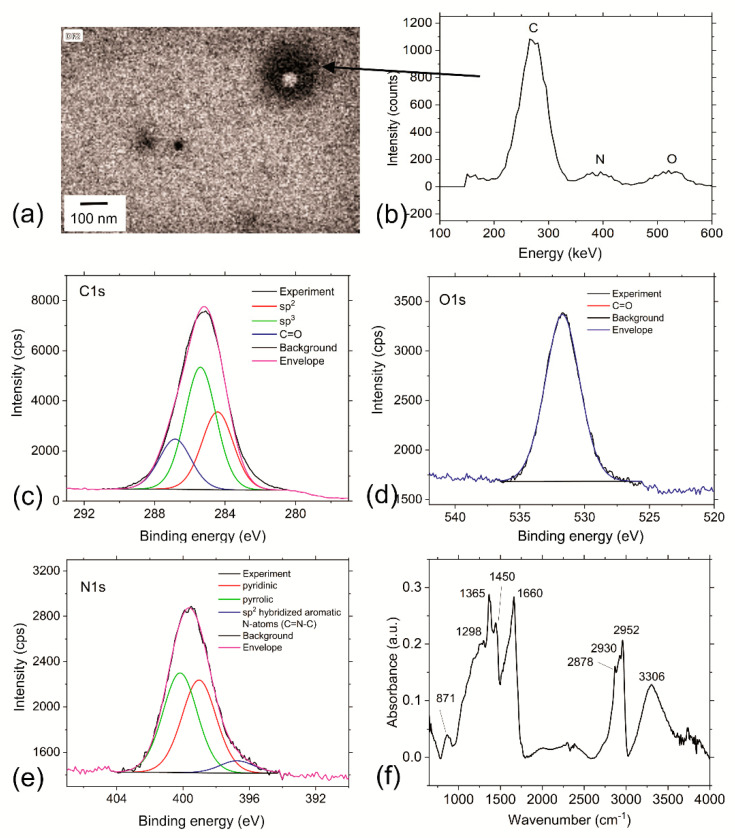
(**a**) TEM image of RF-CPDs selected for chemical analysis. (**b**) HAADF chemical analysis of the RF-CPDs on TEM support grid. (**c**) Deconvoluted XPS C1s peak of the RF-CPDs. (**d**) Deconvoluted XPS O1s peak of the RF-CPDs. (**e**) Deconvoluted XPS N1s peak of the RF-CPDs. (**f**) FTIR spectrum of the RF-CPD nanoparticles.

**Figure 4 nanomaterials-12-04070-f004:**
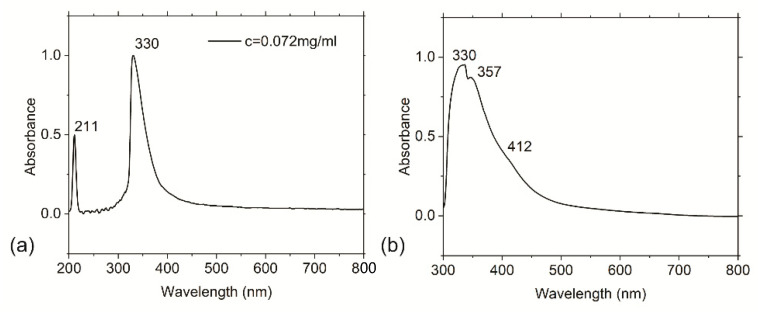
UV-Vis spectra of (**a**) RF-CPD nanoparticles and (**b**) RF-CPD/PU composites.

**Figure 5 nanomaterials-12-04070-f005:**
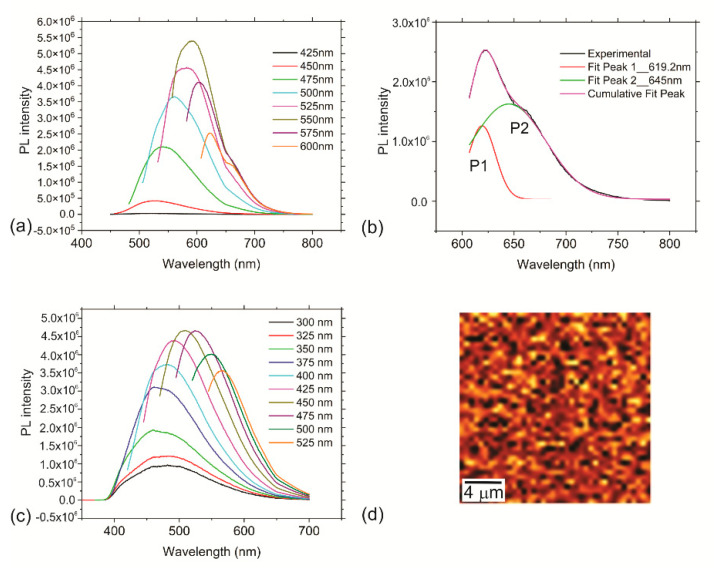
(**a**) PL spectra of the RF-CPD nanoparticles. (**b**) Fitted PL spectrum of the RF-CPD nanoparticles at a 600 nm excitation wavelength. (**c**) PL spectra of the RF-CPD/PU composites. (**d**) CRM mapping of the RF-CPD/PU composites.

**Figure 6 nanomaterials-12-04070-f006:**
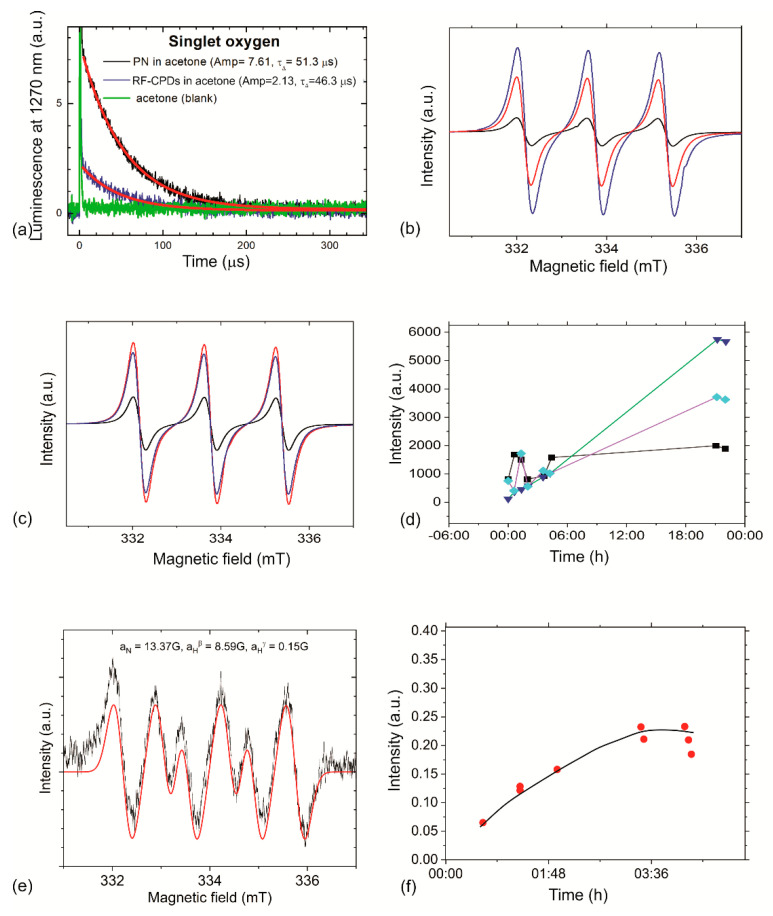
(**a**) Laser-excited singlet oxygen luminescence of the RF-CPDs (blue curve) and PN (black curve) vs. time in oxygen atmosphere. The red line is an exponential fit of experimental data with the calculated lifetime of singlet oxygen, τ_Δ_. (**b**) The EPR signal intensity of the RF-CPDs (red curve) and nicotinic acid (blue curve). Both colloids were dissolved in acetone. (**c**) The EPR signal intensity of the RF-CPD/PU composites (red curve) and C_60_/PU composites (blue curve). (**d**) The kinetics of singlet oxygen production vs. time (control—black square), C_60_/PU (cyan square), and RF-CPD/PU composite (dark blue triangle). (**e**) Production of superoxide anions by the RF-CPD/PU composites. (**f**) Kinetics of the superoxide anions production by the RF-CPDs vs. time.

**Figure 7 nanomaterials-12-04070-f007:**
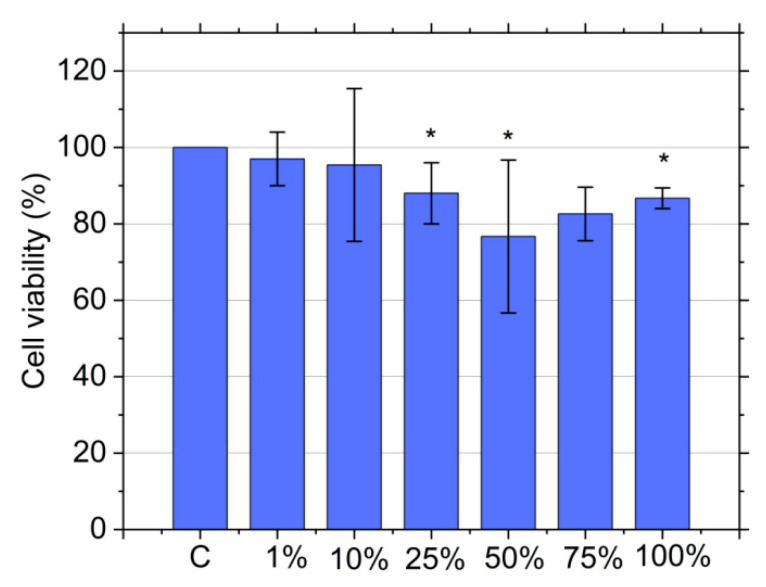
Cytotoxicity of the RF-CPD/PU composites against MRC-5 cells. MRC-5 cells were treated for 24 h with RF-CPD/PU composites in an incubated medium (1, 10, 25, 50, and 100%). Cell viability was expressed as the percentage of absorbance relative to the untreated cells for the RF-CPD/PU composites, which was set at 100%. Data are presented as the mean ± SD of at least three independent experiments. Asterisk indicate statistical significance (* *p* < 0.05). SD—standard deviation.

**Table 1 nanomaterials-12-04070-t001:** Calculated photophysical parameters of the RF-CPDs.

Compound	Solvent	Amp	τ_Δ_ (μs)	τ_Δ_ (μs) [63]	Φ_Δ_
Phenalenone (PN)	Acetone	7.61	51.3	30–65	1.00 [44]
RF-CPDs	Acetone	2.13	46.3		0.28

**Table 2 nanomaterials-12-04070-t002:** Antibacterial activity of the RF-CPD/PU composites against *E. coli* and *S. aureus*.

BL (min)	*E. coli*		*S. aureus*	
	N * (CFU/cm^2^)	R	N (CFU/cm^2^)	R
30	1.7 × 10^3^	1.4	1.2 × 10^4^	0.3
60	0.9 × 10^3^	1.6	3.8 × 10^3^	0.8
90	0.3 × 10^3^	2.1	1.8 × 10^3^	1.1

* N—number of viable bacteria recovered for test specimen; R—antibacterial activity of tested material; Untreated *E. coli*: N = 3.9 × 10^4^ (U_t_ = 4.59); Untreated *S. aureus*: N = 2.2 × 10^4^ (U_t_ = 4.34).

## Data Availability

Not applicable.

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
