# Peer review of "Highly Efficient Antibacterial Polymer Composites Based on Hydrophobic Riboflavin Carbon Polymerized Dots"

_nanomaterials, 2022, doi:10.3390/nano12224070_

Round 1

Reviewer 1 Report

This paper describes synthesis, characterization and biological testing of novel antibacterial polymer composites based on riboflavin-containing carbonized polymer dots encapsulated into polyurethane films. The above composites demonstrate effective ROS generation (both through the energy and electron transfer mechanisms), high antibacterial activity against gram-positive and gram-negative bacteria and low dark cytotoxicity. Hence, such hybrid materials can be applied as self-cleaning coatings capable of microbial photoinactivation under the low-power visible light irradiation.

The work is performed at a high experimental level, and the results obtained seem to be reliable and self-consistent. Conclusions are fully supported by the data obtained, and a comprehensive discussion reveals the mechanisms underlying the effects observed. So, this paper is expected to be of great interest for biomedical and materials scientists and it is worth publishing in Nanomaterials without any serious modifications except for proofreading and correction of the typos throughout the text.

Author Response

Answers to referee 1

Thank you for careful reading of our manuscript. We appreciate your comments.

Best regards

Zoran Marković

Reviewer 2 Report

This is a thorough and profound study on polyurethane-based materials comprising riboflavin and carbonized polymer dots. The materials were characterized with a number of different techniques, and investigations on photoactivity and antimicrobial properties were performed. Thus many aspects are addressed which renders this study a comprehensive work of particular value.  The manuscript is well structured and the text is illustrated with valuable figures of good quality. Only the large number of acronyms makes reading somewhat difficult. After implementation of minor remarks as outlined below I will recommend this work for publication in nanomaterials.

p. 3, Section 2.1., line 2 – 3 (total line 126 – 127): The pore size of the Nylon membrane filter should be indicated also here (although it is mentioned later in Section 2.2.).

p. 3, Section 2.1., line 3 (total line 127): The acronym TMP should be defined here.

p. 3, Section 2.1., line 4 (total line 128): The acronym DMPO should be included in brackets.

p. 3, Section 2.1., line 5 (total line 129): As there is a plethora of different polyurethanes, more information on the nature of the used polyurethane should be provided (chemical composition, molecular weight).

p. 4, section 2.4., 2nd paragraph: In the course of the description of the XPS, it should be indicated how the energy was calibrated (284.8 eV to adventitious carbon?).

p. 5 -6, section 2.8.: It is not really evident from the description of the contact angles if the advancing or the receding contact angle was measured. This should be specified clearly.

p. 6, section 3.1., line 1 (total line 270): It is not evident from the section Materials (section 2.) how the concentration of the RF-CPDs was determined by thermal gravimetry.

p. 6, section 3.2., line 3 (total line 276): It seems to me that the acronym RB was not defined before – all in all there are too many acronyms and it is difficult to keep them all in mind (as already mentioned above).

p. 7, section 3.2., line 4 and line line 9 (total line 277 and line 282): The number of decimal places in the average values of the core diameters (two decimal place) and their deviations (no decimal place) is different. This does not make any sense. Both the average values and their deviations should have the same number of decimal places (no decimal place or maximum one decimal place). Also it should be indicated which confidence level is used for the deviations (95% confidence level?).

p. 7. Section 3.1., last paragraph, line 3 (total line 316): Both the peaks at 2952 cm-1 and 2930 cm-1 are attributed to C-H stretching vibrations of CH3 groups. However, the frequency of 2930 cm-1 is rather characteristic for CH2 groups.

Author Response

Answers to referee 2

We read all referee’s comments carefully. All comments have been accepted and the manuscript has been corrected as requested. Thank you for careful reading of our manuscript. We appreciate your comments.

  1. 3, Section 2.1., line 2 – 3 (total line 126 – 127): The pore size of the Nylon membrane filter should be indicated also here (although it is mentioned later in Section 2.2.).

The pore size of membrane filter has been inserted in the 2.1. section.

  1. 3, Section 2.1., line 3 (total line 127): The acronym TMP should be defined here.

The full name of used chemical has been inserted in the 2.1 section.

  1. 3, Section 2.1., line 4 (total line 128): The acronym DMPO should be included in brackets.

The acronym DMPO has been inserted in the brackets.

  1. 3, Section 2.1., line 5 (total line 129): As there is a plethora of different polyurethanes, more information on the nature of the used polyurethane should be provided (chemical composition, molecular weight).

Polyurethane used is commercial type, donated by American Polyfilm Co. Only data provided by the producers is the thickness, shore and purity. In our previous research, we investigated the chemical composition of this medical grade polyurethane (M. Kovačova et al. ACS Biomater. Sci. Eng. 2018;4:3983).

  1. 4, section 2.4., 2nd paragraph: In the course of the description of the XPS, it should be indicated how the energy was calibrated (284.8 eV to adventitious carbon?).

The sentence related to the binding energy axis calibration was added in the experimental part. Page 4, section 2.4.

  1. 5 -6, section 2.8.: It is not really evident from the description of the contact angles if the advancing or the receding contact angle was measured. This should be specified clearly.

The instrument used has only option for measurements of standard static drop contact angle. Instrument does not have possibility for determination of dynamic contact angles (advancing or receding contact angle)-webpage of instrument producer is advex-instruments.cz.

  1. 6, section 3.1., line 1 (total line 270): It is not evident from the section Materials (section 2.) how the concentration of the RF-CPDs was determined by thermal gravimetry.

The detailed information has been added to manuscript. (page 4, section 2.2)

  1. 6, section 3.2., line 3 (total line 276): It seems to me that the acronym RB was not defined before – all in all there are too many acronyms and it is difficult to keep them all in mind (as already mentioned above).

Sorry, it was mistake. The real acronym for riboflavin is RF not RB. Now, the mistake has been corrected in the section 3.2.

  1. 7, section 3.2., line 4 and line line 9 (total line 277 and line 282): The number of decimal places in the average values of the core diameters (two decimal place) and their deviations (no decimal place) is different. This does not make any sense. Both the average values and their deviations should have the same number of decimal places (no decimal place or maximum one decimal place). Also it should be indicated which confidence level is used for the deviations (95% confidence level?).

We conducted again the statistics on 20 micrographs and calculated the following: Mean value of diameter of average core diameter is 35.2 nm. Standard deviation is 2.1 nm. Mean value of the average diameter of these dots is 57.4 nm. Standard deviation is 3.1 nm (Figure 2c). Results in the manuscript are presented in form 35.2±2.1 nm and 57.4±3.1 nm. Standard deviation means that all values of diameters (core, whole dot) fall within this range.

  1. 7. Section 3.1., last paragraph, line 3 (total line 316): Both the peaks at 2952 cm-1 and 2930 cm-1 are attributed to C-H stretching vibrations of CH3 groups. However, the frequency of 2930 cm-1 is rather characteristic for CH2 groups.

We have accepted the reviewer’s comment and corrected this detail in the manuscript. Page 8, section 3.3

Best regards

Zoran Marković

Reviewer 3 Report

Title: Highly efficient antibacterial polymer composites based on hydrophobic riboflavin carbon polymerized dots.

##Overall comments

The paper describes the preparation of an antimicrobial coating using riboflavin carbon dots and polyurethane. The presentation of the paper needs to be sound. The objective and utility need to be clarified. The figures could be more precise. More experiments are required to prove the formation of riboflavin carbon dots. For these reasons, I cannot recommend the paper for publication in this journal. Sorry for it.

##Comments on the title, Abstract, and References

1. The title is OK,

2. "Detailed structural, optical, antimicrobial, and cytotoxic investigations of these composites have been conducted" please elaborate.

3. Please include the utility of this material in the abstract. 

##Comments on the introduction section

5. line 56-57: "However, there are several disadvantages of both chemical and chemical-free approaches." Please include the disadvantages.

6. Line 66: Please write the wavelength of the blue light.

The objective needs to be clarified.

##Comments on Experimental

7. Line 63" Riboflavin (RF), also known as vitamin B2, belongs to the class of water-soluble vitamins. It is not soluble in acetone or chloroform" However, in line: 132, the authors used acetone to prepare carbonized polymer dots. Which one, right? 

8. What is the role of ethylenediamine? is it soluble in acetone?

9. Presentation of figures could be better. Suggest redrawing each figure with high resolution and replacing the old one.

10. objective and utility of the findings need to be clarified.

Author Response

Answers to referee 3

We read all referee’s comments carefully. All comments have been accepted and the manuscript has been corrected as requested. Thank you for careful reading of our manuscript. We appreciate your comments.

  1. The title is OK

Thank you.

  1. "Detailed structural, optical, antimicrobial, and cytotoxic investigations of these composites have been conducted" please elaborate.

In the manuscript we have conducted structural analysis of RF-CPDs and RF-CPDs composites. Structural analysis includes the investigation of size and height of RF-CPDs, surface roughness of RF-CPDs composite and determination of chemical composition by elementary analysis, XPS and FTIR measurements. Antimicrobial testing includes the determination of possible antimicrobial activity of these composites against two bacteria strains: Escherichia coli (Gram negative) and Staphylococcus aureus (Gram positive). According to different studies these bacteria can be found on surfaces in hospitals and healthcare facilities. Cytotoxicity measurement should provide information about the toxicity of composites applied on surfaces.

  1. Please include the utility of this material in the abstract.

We have included the utility of this material in the abstract.

##Comments on the introduction section

  1. line 56-57: "However, there are several disadvantages of both chemical and chemical-free approaches." Please include the disadvantages.

We have included the disadvantages in the text (Introduction, page 1)

  1. Line 66: Please write the wavelength of the blue light.

The wavelength of blue light used is 455 nm (ref 20).

  1. The objective needs to be clarified.

The main objectives of this study are the following: design of photodynamic antimicrobial surface based on riboflavin, its characterization and potential biomedical application. Page 1, introduction section

##Comments on Experimental

  1. Line 63” Riboflavin (RF), also known as vitamin B2, belongs to the class of water-soluble vitamins. It is not soluble in acetone or chloroform” However, in line: 132, the authors used acetone to prepare carbonized polymer dots. Which one, right?

Riboflavin is not soluble in acetone at room temperature. CPDs are prepared at 180oC in the hydrothermal reactor. Obtained product is highly dispersible in acetone.

  1. What is the role of ethylenediamine? is it soluble in acetone?

Ethylenediamine is well-known stabilizer of riboflavin. It promotes dispersion of riboflavin at room temperature. It is soluble in acetone. At elevated temperatures it reacts with riboflavin and produce stabile colloidal nanoparticles in acetone.

  1. Presentation of figures could be better. Suggest redrawing each figure with high resolution and replacing the old one.

We have improved the quality of figures.

  1. objective and utility of the findings need to be clarified.

Objectives and utility of the study are clarified in the abstract, introduction and conclusion.

Best regards

Zoran Marković

Round 2

Reviewer 3 Report

Thanks for the improvement in the resolution of the figures. However, the text in Figure 1 is not visible.

Ethylenediamine is a strong base soluble in water, alcohol, and ether. How is it soluble in acetone? Besides, ethylenediamine is widely used in fungicides, chelating agents such as EDTA, resins, textiles, lubricants, and as a solvent and emulsifier.

Besides, riboflavin could be stabilized at 96.2%  by the disodium ethylenediamine( References:  https://doi.org/10.3762/bjoc.10.208 ).

Line 139-141: "Carbonized polymer dots (RF-CPDs) were prepared by a one-step hydrothermal method" Authors used the hydrothermal technique but without water. Is it hydrothermal? 

Therefore, the findings of the results are questionable. 

Sorry for it. 

Author Response

Answers to referee 3

  1. Thanks for the improvement in the resolution of the figures. However, the text in Figure 1 is not visible.

Figure 1 has been improved, i.e. the font of the letters was enlarged.

  1. Ethylene diamine is a strong base soluble in water, alcohol, and ether. How is it soluble in acetone? Besides, ethylenediamine is widely used in fungicides, chelating agents such as EDTA, resins, textiles, lubricants, and as a solvent and emulsifier.

Answer: General literature (Ethylenediamine 107-15-3 | TCI AMERICA (tcichemicals.com)) claims that Ethylenediamine is miscible with water and soluble in alcohol, ether and heptane.

In the paper “Reaction between ethylenediamine and acetone on a platinum(II) complex. Crystal structure of chloro(ethylenediamine)(tributylphosphine)platinum(1+) chloro(N-isopropylideneethylenediamine)(tributylphosphine)platinum(1+) dichloride.acetone” Inorg. Chem. 1988, 27, 21, 3866–3868, authors used in the experiment solution of ethylenediamine (0.1mmol) in acetone (0.5ml).

In our experiment, at first riboflavin was mixed in acetone for 5 minutes which resulted with opaque dispersion. Upon dropwise addition of ethylenediamine, dispersion got transparent dark orange colour. That liquid was used for further experiment.  (This paragraph was added in the section 2.2, page 3)

  1. Besides, riboflavin could be stabilized at 96.2% by the disodium ethylenediamine (References: https://doi.org/10.3762/bjoc.10.208 ).

Answer: Thank you for your suggestion. We will try to produce carbon dots by usage of disodium ethylenediamine.

  1. Line 139-141: "Carbonized polymer dots (RF-CPDs) were prepared by a one-step hydrothermal method" Authors used the hydrothermal technique but without water. Is it hydrothermal?

Answer:  In the numerous papers carbon quantum dots were produced by hydrothermal method with solvents such as DMF( https://doi.org/10.1016/j.biortech.2021.126143, https://doi.org/10.1016/j.cej.2021.131653 , https://doi.org/10.1002/bio.3698 , https://doi.org/10.1039/C8RA01085D ), ethanol (https://doi.org/10.1016/j.colsurfa.2022.129261, https://doi.org/10.1016/j.indcrop.2022.114957 , https://doi.org/10.1016/j.jhazmat.2020.124422) or other solvents.  Other rarely used term is solvothermal method although authors use identical reactor for synthesis. Since scientific community is not unified in usage of this term, we used term hydrothermal in this manuscript.

Best regards,

Zoran Marković